# Genetic variation across trophic levels: A test of the correlation between population size and genetic diversity in sympatric desert lizards

Erica M. Rutherford[¤a], Andrew Ontano[¤b], Camille Kantor, Eric J. Routman[iD] *

Department of Biology, San Francisco State University, Holloway Avenue, San Francisco, California, United States of America

¤a Current address: Second Genome, South San Francisco, California, United States of America
¤b Current address: Department of Zoology University of Wisconsin–Madison, Madison, Wisconsin, United States of America
* routman@sfsu.edu

## Abstract

Understanding the causes of genetic variation in real populations has been elusive. Competing theories claim that neutral vs. selective processes have a greater influence on the genetic variation within a population. A key difference among theories is the relationship between population size and genetic diversity. Our study tests this empirically by sampling two species of herbivorous lizards (*Dipsosaurus dorsalis* and *Sauromalus ater*) and two species of carnivorous lizards (*Crotaphytus bicinctores* and *Gambelia wislizenii*) that vary in population size at the same locality, and comparing metrics of genetic diversity. Contrary to neutral expectations, results from four independent loci showed levels of diversity were usually higher for species with smaller population sizes. This suggests that selective processes may be having an important impact on intraspecific diversity in this reptile community, although tests showed little evidence for selection on the loci sequenced for this study. It is also possible that idiosyncratic histories of the focal species may be overriding predictions from simple neutral models. If future studies show that lack of correlation between population size and genetic diversity is common, methods using genetic diversity to estimate population parameters like population size or time to common ancestor should be used with caution, as these estimates are based on neutral theory predictions.

## Introduction

Genetic diversity is essential to the process of evolution, however we are still trying to understand the relative importance of factors that influence genetic diversity in natural populations. At the level of DNA sequence diversity, the *Neutral Theory* predicts that most molecular genetic variation found in populations is selectively neutral, and is shaped primarily through the random processes of mutation and genetic drift [1]. An important prediction of this theory

**Data Availability Statement:** All DNA sequence files are available from the NCBI database (accession numbers MK780237- MK780740).

**Funding:** The authors received no specific funding for this work.

**Competing interests:** The authors have declared that no competing interests exist.

is that the size of a population will have a large impact on its genetic diversity. Genetic drift has a greater effect on smaller populations, leading to a predicted positive correlation between population size and within-population genetic diversity. Empirical evidence from natural populations has been mixed. In some studies, the neutral theory appears to adequately explain observed patterns of genetic diversity [2–4], see [5] for review, while others have found results which do not fit neutral predictions [6–7], see [8] for review, or in which distantly related species with vastly different effective population sizes have a lower than expected difference in genetic diversity [6]. Natural selection is thought to be the most likely explanation for the deviations from neutrality found in [6–8], especially for genetic regions with low recombination.

Alternative theories suggest that natural selection may play a larger role in determining levels of polymorphism through both background (purifying) selection and genetic hitchhiking. Theories such as the *Nearly Neutral Theory* [9] and *Genetic Draft* [10, 11] have been proposed as mechanisms by which purifying or balancing selection may shape patterns of genetic variation in the genome. Because natural selection operates more efficiently in large populations than in small ones these theories predict that, except in very small populations, there will be little or no correlation between population size and within-population genetic diversity. This is true not only for selected loci but also for neutral loci in linkage disequilibrium with selected loci. Therefore, selection is expected to have more extensive effects in mitochondrial DNA, haploid organisms, and regions of low recombination [11, 12], since recombination can separate a mutation with a high selection coefficient from its original allelic background. Genome wide polymorphism data appears to support this theory, suggesting that the effects of selection are greater in species with larger population sizes [13]. The accuracy of the statistical methods used to identify regions affected by selection in the genome remains controversial [5, 14].

Despite the amount of attention the role of population size in population genetics has received, no theory has yet fully explained the patterns of genetic variation seen in natural populations [14–16]. Surprisingly few studies have been conducted on natural populations to test the relationship between population size and genetic diversity. The studies that exist often draw conclusions based on samples taken across a species' range rather than from a single deme [13, 17, 18] thereby confounding within- and between-population effects of evolutionary forces like drift and selection as well as increasing the likelihood that populations have experienced different historical effects. Differences in conditions across the range of a species could also cause large differences in selective pressures. Clearly these factors make it difficult or impossible to untangle patterns of genetic diversity when samples are taken over large geographic distances.

An ideal method for testing neutral expectations is to compare the genetic diversity of taxonomically related species living sympatrically. By focusing on a taxonomically related group, it is possible to reduce many confounding variables that may differ among taxonomically more diverse contrasts, such as potential differences in DNA mutation rates and constraints on locus function. Studying a single area reduces (but does not eliminate) the possibility that the gene pools of the focal species have been responding to radically different events in recent geologic history, or that patterns caused by population subdivision are being conflated with patterns within a deme.

Our research compares genetic diversity in species of lizards that differ in population size. Our previous studies found interspecific differences in levels of genetic diversity in sympatric populations of lizards in the Mojave Desert [19, 20]. Hague and Routman [20] showed that species with smaller population sizes had generally lower genetic diversities than species with larger population sizes. We surveyed genetic variation in two additional species with smaller population sizes from the same locality, and in this paper we compare their DNA sequence diversity to our previous results and expand on the previous datasets. For four focal lizard

species we tested the neutral theory prediction of the positive relationship between genetic diversity and population size using three criteria to rank relative population size: (a) trophic level, (b) habitat specialization, and (c) a combination of published abundance surveys and long term observation of the lizard community at our study site. The above criteria predict that population size, and therefore genetic diversity would rank as follows: carnivore specialist< carnivore generalist< herbivore specialist< herbivore generalist.

## Methods

### Ethics statement

All procedures involving animals in this study followed ethical and legal guidelines. The research protocol was approved by the San Francisco State University Institutional Animal Care and Use Committee (IACUC; animal protocol #A14-06). All collection took place on federally owned land, under National Park Service Scientific Research and Collecting Permit # MOJA-2014/5-SCI-0002 Study #00270 issued to EJR. None of the four focal species are federally listed as threatened or endangered, or listed as species of special concern in the state of California.

### Taxa sampled

The lizard communities of the Mojave Desert of North America make an ideal study system for a study of comparative population genetics. Numerous species are found living sympatrically, occupying many niches within the environment. Many species reach high local abundances while others are much less dense. This study focuses mainly on two members of the family Crotaphytidae (*Crotaphytus bicinctores*, the Great Basin Collared Lizard, and *Gambelia wislizenii*, the Long-nosed Leopard Lizard), and two members of the family Iguanidae (*Dipsosaurus dorsalis*, the Desert Iguana, and *Sauromalus ater*, the Common Chuckwalla) in the Mojave National Preserve, San Bernardino County, California, USA. While relationships among the pleurodont reptiles are not fully resolved, it is clear that Crotaphytidae and Iguanidae are closely related to one another [21, 22]. These four species are all large-bodied lizards [Snout-vent lengths: *C. bicinctores*—8.6–11.2 cm, *G. wislizenii*—8.2-14-6 cm, *D. dorsalis*—10.1–14.6 cm, *S. ater*—12.7–22.8 cm] with long generation times, essentially controlling for factors other than trophic level and habitat niche. The two crotaphytids are predators, and often prey on smaller lizards [23, 24], while the two iguanids are primarily herbivorous [25, 26]. The general expectation that carnivorous species will be less common than herbivorous species of similar size has been supported in this instance by previous observations of the Mojave lizard community [27, 28]; authors' observations). By sampling two species from each trophic level we have added a degree of replication to the study design, although it is confounded with phylogeny in this case. *G. wislizenii* and *D. dorsalis* are desert habitat generalists, while *C. bicinctores* and *S. ater* are found only among rocks. Because each lizard family/trophic level group contains one species in each category, we do not expect habitat niche to be a main determinant of genetic diversity differences between trophic levels. In addition, these four species are not known to differ in characteristics that should affect the relationship between $N_e$ and actual population size, such as tendency to inbreed, and are relatively well matched in body size and generation time.

Some genetic work has already been conducted on these, or closely related species. Phylogeographic studies found very high genetic diversity across the range of *Sauromalus obesus* (= *ater*) [29], and *S. obesus* and *D. dorsalis* [30], which was likely attributable to population subdivision. A phylogeographic study of *G. wislizenii* in the Mojave Desert found high haplotype diversity in a single mitochondrial gene, but this appeared to be based on only a few

individuals collected from three distant localities [31]. A comprehensive phylogenetic study of the family Crotaphytidae sampled 408 individuals, with *C. bicinctores* and *G. wislizenii* heavily represented [32], but made no attempt to quantify intraspecific diversity. Earlier work at this sitefound that the two large iguanids studied here had less genetic diversity than small-bodied species with very high population densities [19,20].

## Population size

One difficulty in testing neutral predictions is that it is the effective population size that is expected to affect diversity, rather than the current census population size. Effective population size ($N_e$) is defined as the number of idealized (usually Wright–Fisher model) individuals needed to account for the observed levels of genetic diversity in the actual population. The relationship between $N_e$ and the actual population size depends on many factors, only one of which is the actual number of individuals, and in practice $N_e$ is usually estimated from empirical estimates of genetic diversity ($\theta$) and the mutation rate ($\mu$) ($N_e = \theta/4\mu$ for autosomal genes). However, this equation *assumes* neutrality and cannot be used in studies testing whether genetic diversity and population size are correlated.

In this study, we use a combination of trophic level expectation and long-term observation of relative abundance as a proxy for relative $N_e$. Ecological theory predicts that, all else being equal, herbivores will be more abundant than carnivores of equal body size and that this relationship is likely to be stable over time[33,34]. Population census [27,28,35] and over 20 years of field work at the study site by one of us (Routman) supports the proposition that these carnivorous species are much less abundant than the herbivorous species. Indeed, for the carnivorous species it took over four field seasons to collect samples that were similar in size to those of the herbivorous species, which were collected in one to two field seasons. This difference in time to complete the sample is an underestimate of the relative population sizes, because most of the samples of the herbivorous species were collected at the start of our work on genetic diversity in the Mojave National Preserve. At that time we were simultaneously searching for 16 different species found in several different habitats, whereas the collection of most of the carnivorous species were later in our overall study, when sampling of most of the common species was completed. Thus, in the later part of the study we were focusing our efforts mainly on the carnivorous species.

Support for the idea that, within trophic level, the rock specialists should have lower population sizes than those of habitat generalists because of less available habitat can be inferred from Fig 1. The fact that rocky outcrops cover much less area than the intervening sandy regions, combined with the fact that the habitat generalist also use the rocky habitats (albeit in lower density) essentially insures that, within trophic level, the habitat generalist will have a larger population size.

Some survey data from the Mojave National Preserv supports our ranking of population sizes in these species. Data from a can trap grid [35], collected monthly June 1991-May 1993, January 2000-December 2001, and January 2008-June 2018, yielded the following capture numbers for the 4 focal species: *D. dorsalis*—68, *G. wislizenii*– 4, *C. bicinctores*– 2, *S. ater*—3. These number reflect our rankings with the exception of *S. ater*, which has a capture frequency closer to that of the two carnivores. However, *S. ater* are rarely caught in can trap studies, because their preferred habitat is large boulder fields and rocky cliffs, where can traps cannot be used because the cans require burying.

## Collection and sequencing

Collection took place primarily in the Cima Volcanic Field (referred to as Lava below) of the Mojave National Preserve, CA. (Fig 1). This locality was sampled in our previous studies

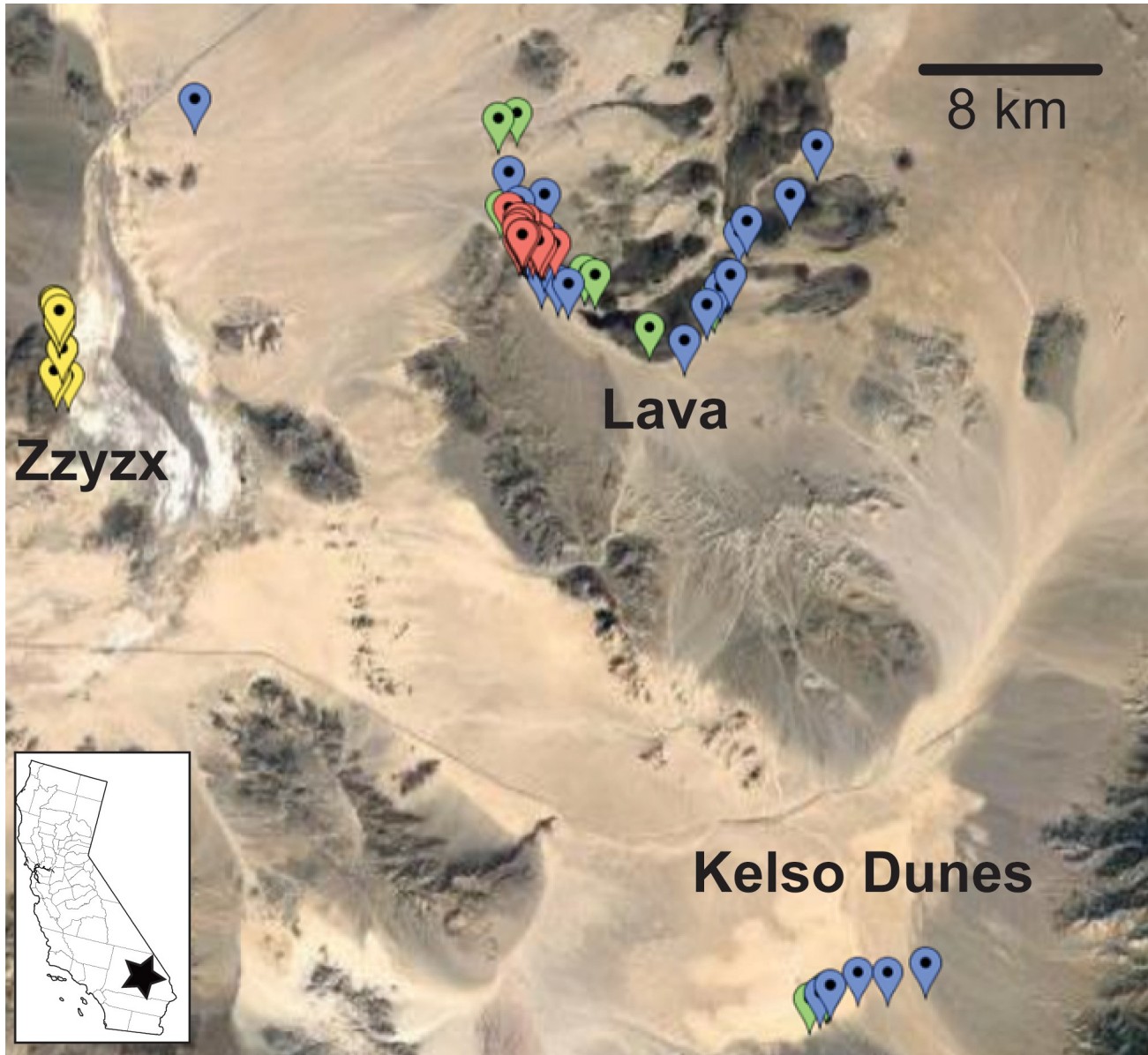

**Fig 1. Collection sites in the Mojave National Preserve for four species of lizard.** Markers indicate individual lizard collection localities, separated by species as follows: Yellow: *Crotaphytus bicinctores*; Blue: *Gambelia wislizenii*; Green: *Dipsosaurus dorsalis;* Red: *Sauromalus ater*. Because of the map scale, there is considerable marker overlap, e.g. *C. bicinctores* markers at the Lava site are not visible.

[19,20], allowing for an accurate comparison between datasets. Due to the low abundance of the crotaphytid species, we collected some individuals from nearby desert areas. All of the *S. ater* samples(n = 37), along with 36 *D. dorsalis*, 11 *C. bicinctores*, and 27 *G. wislizenii* were collected at the Cima Volcanic Field. In order to increase sample sizes, two nearby areas were used as secondary collection sites. 21 *C. bicinctores* were collected from the nearby Zzyzx Road (23 km from the Cima Volcanic Field). Because the Zzyzx Rd. site has a low density of *G. wislizenii*, we also collected at the area near Kelso Dunes (35 km from the Cima Volcanic Field), which yielded 5 *G. wislizenii* and 4 *D. dorsalis*.

Collection took place over four years (2012–2015). Iguanid samples previously collected and sequenced by Hague and Routman [20] were used for this study, and supplemented by

additional individuals of both species to increase sample size. Exact locality data for each specimen can be found in S1 Table. We collected lizards of the target species using slip-knot nooses, and sampled a small piece of tissue (0.5 cm) from the tail of each individual, which was then released. Samples were preserved in 95% ethanol in the field, and subsequently frozen.

We sequenced protein coding regions of mitochondrial cytochrome b (*cytb*) and three autosomal genes: melanocortin 1 receptor gene (*MC1R*), recombination activating gene 1 (*RAG1*), and caspase recruitment domain gene 4 (*CARD4*). *Cytb* is commonly used for phylogenetic studies of mitochondrial DNA in vertebrates, and because the mitochondrial genome is non-recombining, should be more affected by genetic hitchhiking than the autosomal genes. We sequenced approximately the same segment of each gene for each of the four species to control for possible differences in variability among regions of the gene.

We used standard tissue extraction methods to prepare DNA from the samples for sequencing (Quick-gDNA™ MiniPrep kit; Zymo Research). PCR was carried out in Accupower Pyro-HotStart Taq PCR Premix tubes (Bioneer, Inc.) to amplify the selected genes (See S1 File for primers and PCR protocols used). After purification using ExoSAP-IT® (Affymetrix), we sent the resulting PCR product to Elim Biopharmaceuticals (Hayward, CA) for Sanger sequencing in both directions. PCR primers were also used for sequencing except for a few locus-species combinations for which we had to design sequencing primers. We aligned sequence reads and evaluated sequences visually for heterozygosity and accuracy of sequence data using Geneious version 7.1.5 (http://www.geneious.com) [36].

## Data analysis

We sequenced a segment of mitochondrial *cytb* ranging in size from 610–872 base pairs (Table 1). The ends of the alignment for each species were manually trimmed to include only base pairs present in ≥95% of the aligned individuals. Only the ends of some sequences had uncalled bases, involving < 10 bases. Because Arlequin can give nonsensical results for some analyses when missing bases are present at the ends of sequences, we manually replaced any uncalled base with the base present in that position in other individuals having the same sequence for the scored bases (according to its documentation, this is the default assumption built into Arlequin when missing data is incorporated into the analysis). There were no cases in which polymorphism in these regions caused ambiguity regarding which base should be inserted.

We used the programs PHASE [37,38] and seqPHASE [39] to reconstruct haplotypes from the autosomal genotype data. Haplotypes determined by cloning were available for some heterozygous individuals of *D. dorsalis* (*RAG1* and *MC1R*) and *S. ater* (*MC1R*) [20], and these were used as known haplotypes for PHASE analysis of these species. The best supported haplotype pair was chosen for each individual from the most highly resolved PHASE runs and used for further analysis. As it was not possible to find a best haplotype pair with 90% or greater confidence for each individual, we tested the potential impact of choosing incorrect pairs on our final results. We generated simulated datasets in R [40] using a script that randomly chooses one of the possible haplotype pairs from the ".out_pairs" file for each individual, without taking into account the probability of each haplotype pair being correct (S2 File). A set of 1000 random datasets were generated for each locus-species combination. Batch analyses of these simulated datasets were compared with the results from the analysis using the best supported haplotype pairs.

We used Arlequin (version 3.5.1.2) [41] for the analysis of population level diversity. For each species, we calculated standard measures of population genetic diversity, including haplotype diversity (h), Nei's θ ($\theta_\pi$, or π), and Watterson's θ ($\theta_S$) [42,43]. For all loci, pairwise

**Table 1. Population genetics summary statistics, part 1.**

| Species | Gene | N | # BP | # Variable Sites | # Hap. | Kn:Ks | Kn/Ks ratio |
|---|---|---|---|---|---|---|---|
| *Crotaphytus bicinctores* | *cytb* | 32 | 872 | 37 | 7 | 7:30 | 0.23 |
| | *CARD4* | 64 | 799 | 6 | 7 | 2:4 | 0.50 |
| | *MC1R* | 62 | 642 | 16 | 24 | 5:11 | 0.45 |
| | *RAG1* | 64 | 774 | 10 | 11 | 6:4 | 1.50 |
| *Gambelia wislizenii* | *cytb* | 33 | 610 | 20 | 13 | 3:17 | 0.18 |
| | *CARD4* | 66 | 787 | 15 | 13 | 5:10 | 0.50 |
| | *MC1R* | 66 | 741 | 17 | 22 | 4:13 | 0.31 |
| | *RAG1* | 66 | 804 | 20 | 19 | 11:9 | 1.22 |
| *Dipsosaurus dorsalis* | *cytb* | 40 | 844 | 1 | 2 | 1:0 | 1.00 |
| | *CARD4* | 80 | 789 | 3 | 4 | 1:2 | 0.05 |
| | *MC1R* | 80 | 755 | 3 | 4 | 0:3 | 0.00 |
| | *RAG1* | 80 | 1028 | 11 | 13 | 9:2 | 4.50 |
| *Sauromalus ater* | *cytb* | 37 | 848 | 13 | 5 | 4:9 | 0.44 |
| | *CARD4* | 74 | 807 | 6 | 8 | 3:3 | 1.00 |
| | *MC1R* | 74 | 813 | 7 | 6 | 1:6 | 0.17 |
| | *RAG1* | 74 | 1049 | 9 | 7 | 3:6 | 0.50 |

Abbreviations: cytb, cytochrome b; CARD4, caspase recruitment domain gene 4; MC1R, melanocortin 1 receptor gene; RAG1, recombination activating gene 1; N, number of chromosomes sampled; # BP, base pairs; # Hap., number of unique haplotypes; Syn. Mut., synonymous mutations; Nonsyn. Mut., nonsynonymous mutations.

differences between species in genetic diversity measures were considered significant if the 95% confidence intervals of the two estimates did not overlap by more than half of a one-sided error bar [44]. We used Tajima's D to test for deviation from a model assuming neutrality and constant population size [43]. Results were checked in DnaSP version 5.10.01 [45], due to some miscalculations we found in the Arlequin results (see below). Pairwise mismatch distributions were also calculated in Arlequin to test for population expansion. We calculated the ratio of nonsynonymous to synonymous changes for each locus-species combination.

Due to the low observed population sizes of some species, we collected individuals over a wider geographic area than had been used for more abundant sympatric species in order to get a sufficiently large sample size. To account for between-location genetic variation, we tested for population structure within the sampling area. Individuals collected from the same localities were assigned to putative separate subpopulations. We calculated $F_{ST}$ ([46]; eqn. 9) using estimators of heterozygosity adjusted for sample size [47], and used Arlequin to find $\Phi_{ST}$ values between the subpopulations [41]. Due to the high levels of diversity in these lizards, we compared $F_{ST}$ and $\Phi_{ST}$ with Jost's D [48], which estimates the level of differentiation among subpopulations independent of genetic diversity within populations. We calculated Jost's D in R, using eqn. 12 from [48] (S3 File).

## Results

Sample sizes were as follows: 32 *C. bicinctores*, 40 *D. dorsalis*, 33 *G. wislizenii*, and 37 *S. ater* (= 64–80 copies of each autosomal locus). Grouped by trophic level, 65 individuals of carnivorous species and 77 individuals of herbivorous species were included in the sample. Population genetics descriptive statistics are reported in Tables 1 and 2 and the rank order of diversity for all measures is found in Table 3.

**Table 2. Population genetics summary statistics, part 2.**

| Species | Gene | N | Haplotype Diversity | $\theta_S$ | $\theta_\pi$ |
|---|---|---|---|---|---|
| *Crotaphytus bicinctores* | *cytb* | 32 | 0.750 (±0.131) | 10.536 (±7.004) | 8.046 (±8.134) |
| | *CARD4* | 64 | 0.487 (±0.139) | 1.588 (±1.472) | 0.877 (±1.460) |
| | *MC1R* | 62 | 0.936 (±0.029) | 5.307 (±3.706) | 4.491 (±5.199) |
| | *RAG1* | 64 | 0.703 (±0.108) | 2.733 (±2.150) | 1.337 (±1.965) |
| *Gambelia wislizenii* | *cytb* | 33 | 0.866 (±0.076) | 8.079 (±5.813) | 7.861 (±8.589) |
| | *CARD4* | 66 | 0.721 (±0.098) | 4.005 (±2.822) | 3.139 (±3.733) |
| | *MC1R* | 66 | 0.922 (±0.031) | 4.821 (±3.301) | 3.792 (±4.407) |
| | *RAG1* | 66 | 0.909 (±0.031) | 5.226 (±3.454) | 2.699 (±3.293) |
| *Dipsosaurus dorsalis* | *cytb* | 40 | 0.450 (±0.106) | 0.278 (±0.546) | 0.533 (±1.052) |
| | *CARD4* | 80 | 0.561 (±0.094) | 0.768 (±0.922) | 1.052 (±1.649) |
| | *MC1R* | 80 | 0.489 (±0.094) | 0.803 (±0.963) | 0.750 (±1.340) |
| | *RAG1* | 80 | 0.769 (±0.065) | 2.161 (±1.626) | 3.291 (±3.699) |
| *Sauromalus ater* | *cytb* | 37 | 0.590 (±0.086) | 3.672 (±2.836) | 3.517 (±4.096) |
| | *CARD4* | 74 | 0.502 (±0.125) | 1.525 (±1.404) | 1.170 (±1.763) |
| | *MC1R* | 74 | 0.646 (±0.071) | 1.766 (±1.541) | 1.066 (±1.649) |
| | *RAG1* | 74 | 0.737 (±0.055) | 1.760 (±1.414) | 1.812 (±2.270) |

95% confidence interval in parentheses. All θ values are per base and multiplied by 1000 for clarity. Abbreviations: cytb, cytochrome b; CARD4, caspase recruitment domain gene 4; MC1R, melanocortin 1 receptor gene; RAG1, recombination activating gene 1; N, number of chromosomes sampled, θS Watterson's estimator of θ, $\theta_\pi$, Nei's estimator of θ.

## Mitochondrial DNA

The rank order of haplotype diversities was *G. wislizenii > C. bicinctores > S. ater > D. dorsalis*, (Table 2, Fig 2) nearly the reverse of the expectation of Neutral Theory based on population size rank. Haplotype diversity of *G. wislizenii* was significantly greater than that of all other species, while *C. bicinctores* diversity was significantly greater than that of *S. dorsalis* but not *S. ater*.

Watterson's and Nei's θ showed patterns similar to that of haplotype diversity for *cytb* (Tables 2 and 3). *C. bicinctores* and *G. wislizenii* had the highest θ values but were statistically indistinguishable. Relative values of Nei's θ were similar. However, due to the higher standard error associated with this estimator, no significant pairwise differences were found.

Tajima's D was not significantly different from 0 for any species (α = 0.05), suggesting that a stable population size neutral model is appropriate for this locus (Table 4). Conversely, raggedness index values were significant at an alpha level of 0.05 only for *S. ater*, suggesting that *cytb* in the other species have expanding populations. The potential cause for the discrepancy between Tajima's D and the pairwise mismatch results is discussed in the following section.

The ratio of synonymous to nonsynonymous changes was high in every species except *D. dorsalis*, in which the only variable site had a nonsynonymous mutation (Table 1). Across all four species, 21.1% of base changes in *cytb* were nonsynonymous.

## Autosomal DNA

The length of the gene segment sequenced ranged from 787–807 base pairs in *CARD4*, 642–813 bp in *MC1R*, and 774–1049 bp in *RAG1*. All genetic diversity measurements for autosomal loci were calculated from the most likely haplotype pairs identified by PHASE. As expected, PHASE was generally able to find best haplotypes pairs with higher levels of confidence for locus-species combinations with lower genetic diversity (especially lower heterozygosity). The

**Table 3. Rank order comparisons of genetic diversity in four lizard species.**

| | | Haplotype diversity | | | | Nucleotide Diversity $\Theta_S$ | | | |
|---|---|---|---|---|---|---|---|---|---|
| | | *cytb* | *CARD4* | *MC1R* | *RAG1* | *cytb* | *CARD4* | *MC1R* | *RAG1* |
| **Species** | | | | | | | | | |
| | C. b. | 2[a] | 4[a] | 1[ab] | 4[a] | 1[a] | 2 | 1[a] | 2 |
| | G. w. | 1[bc] | 1[abc] | 2[cd] | 1[abc] | 2[b] | 1 | 2[b] | 1[ab] |
| | D. d. | 4[ab] | 2[b] | 4[ace] | 2[b] | 4[abc] | 4 | 4[ab] | 3[a] |
| | S. a. | 3[c] | 3[c] | 3[bde] | 3[c] | 3[c] | 3 | 3 | 4[b] |
| **Pop. size** | | | | | | | | | |
| | Small | 1.5 | 2.5 | 1.5 | 2.5 | 1.5 | 1.5 | 1.5 | 1.5 |
| | Large | 3.5 | 2.5 | 3.5 | 2.5 | 3.5 | 3.5 | 3.5 | 3.5 |
| **Habitat** | | | | | | | | | |
| | Rock | 2.5 | 1.5 | 3.0 | 1.5 | 3.0 | 2.5 | 3.0 | 2.0 |
| | General | 2.5 | 3.5 | 2.0 | 3.5 | 2.0 | 2.5 | 2.0 | 3.0 |

Values in table are species' diversity ranks within a locus, with 1 being the most diverse. Species are listed with the smaller population size species (*C. b.* and *G. w.*) first. Within each column, ranks that share a letter are statistically <u>different</u> ($\alpha \leq 0.05$). None of the $\Theta_N$ estimates were significantly different and are not shown. Lower half of table shows the average ranks of the two small population size species and two larger population size species, respectively. Abbreviations: *cytb*, cytochrome b; *CARD4*, caspase recruitment domain gene 4; *MC1R*, melanocortin 1 receptor gene; *RAG1*, recombination activating gene 1; N, number of chromosomes sampled, $\theta_S$, Watterson's estimator of $\theta$; *C. b.*, *Crotaphytus bicinctores*; *G. w.*, *Gambelia wislizenii*; *D. d.*, *Dipsosaurus dorsalis*; *S. a.*, *Sauromalus ater*.

percentage of individuals with a best haplotype pair supported with 90% or greater confidence ranged from 100% (*C. bicinctores CARD4*, *D. dorsalis CARD4* and *MC1R*) to 52% (*C. bicinctores MC1R* and *G. wislizenii MC1R*). Results from haplotype randomizations showed that using the best pairs from PHASE was unlikely to yield incorrect diversity estimates even if the best pair has low probability (S2 Table). The ranges of haplotype diversity values in the randomized datasets were quite narrow and were within the 95% confidence interval of the best pairs value for each locus in all cases except *MC1R* in *C. bicinctores* and *G. wislizenii* (and even in these cases there was substantial overlap), showing that the choice of haplotype pairs does not have a major effect on the diversity results. Nei's θ values were either the same for every randomized dataset (in eight of eleven locus-species combinations), or formed a narrow range. Watterson's θ values did not vary in any randomized dataset because it is invariant to linkage phase. Therefore, Tajima's D values varied little among randomized datasets.

The average haplotype diversity averaged across species was highest at the locus *RAG1* (0.779) and lowest at *CARD4* (0.568). Significant pairwise differences among some species were found at each autosomal locus (Table 3). In both *CARD4* and *RAG1*, the haplotype diversity of *G. wislizenii* was significantly higher than that of the other three species, which were statistically indistinguishable from one another. At *MC1R*, there was no difference in the very high h values in crotaphytids, but both iguanids were significantly different from one another and significantly lower than the crotaphytids.

For autosomal loci, Watterson's θ values showed significant differences in the many of the pairwise comparisons between species. At *MC1R*, both crotaphytids were significantly more diverse than *D. dorsalis* but not *S. ater*. At *RAG1*, *G. wislizenii* had the highest value and was statistically higher than either iguanid. At *CARD4*, there were no significant pairwise differences between species. As was the case for *cytb*, Nei's θ values showed no significant differences between any species pairs.

Values of Tajima's D were not significantly different from 0 ($\alpha = 0.05$) for any locus-species combination, showing no evidence of selection or changes in population size. The highest absolute value of Tajima's D was found in *G. wislizenii RAG1* (-1.480; p = 0.056).

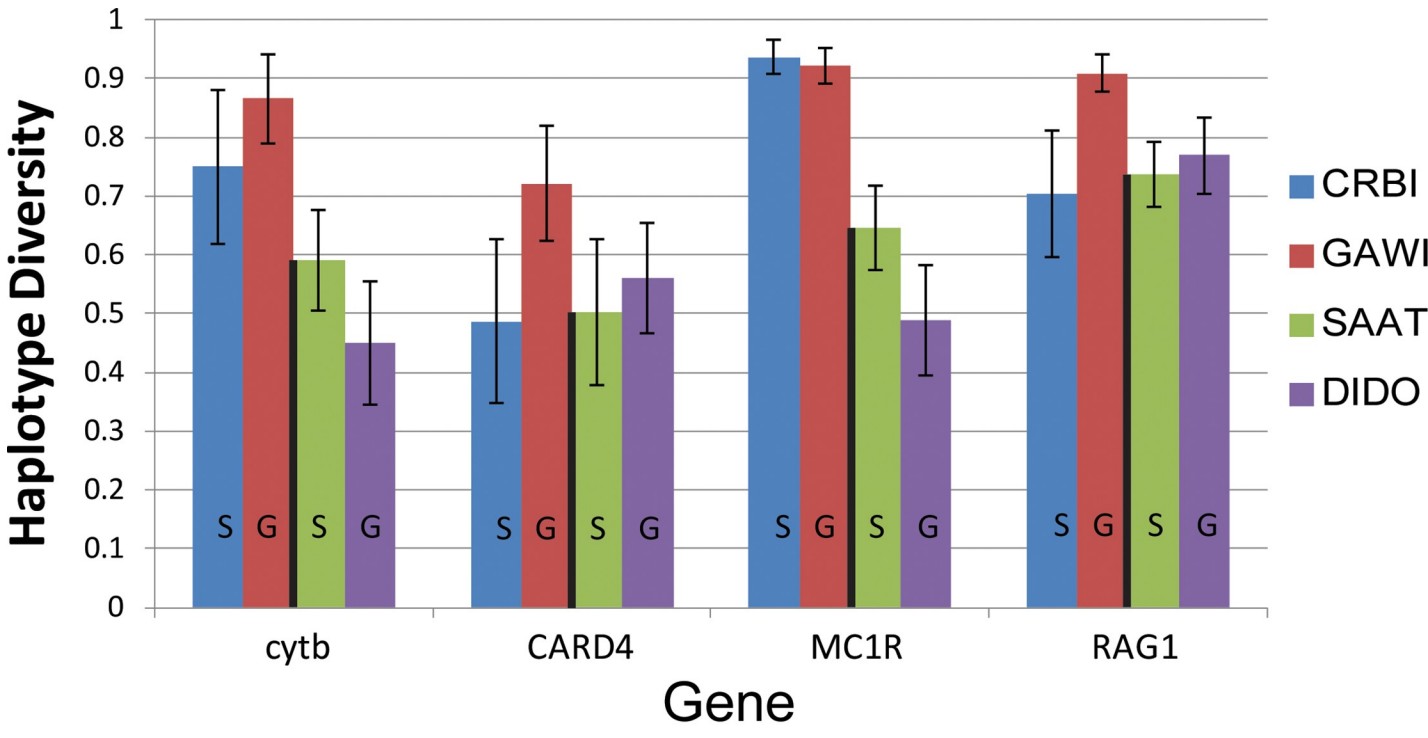

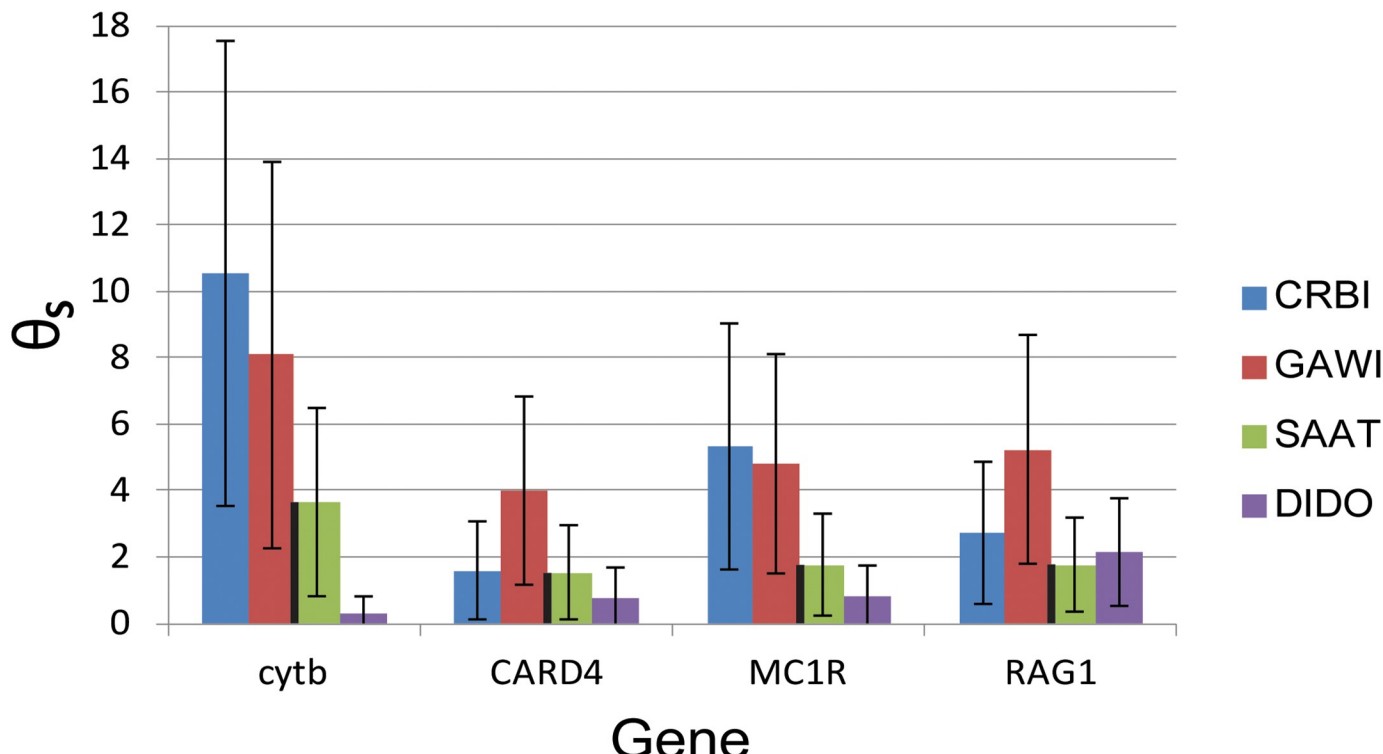

**Fig 2. Haplotype diversity and Watterson's estimate of θ for four species of lizard from the Mojave National Preserve.** Solid black line separates the low population size carnivores (left) from the higher population size herbivores (right). Letters within bars (shown only on the haplotype diversity graph for readability) signify habitat specialist (S) or generalist (G). Gene abbreviations: cytb, cytochrome b; CARD4, caspase recruitment domain gene 4; MC1R, melanocortin 1 receptor gene; RAG1, recombination activating gene 1Species abbreviations: CRBI, *Crotaphytus bicinctores*; GAWI, *Gambelia wislizenii;* DIDO, *Dipsosaurus dorsalis;* SAAT, *Sauromalus ater.* Error bars represent 95% confidence intervals.

**Table 4. Tests of neutral model with constant population size.**

| Species | Gene | D | P-value | R | P-value |
|---|---|---|---|---|---|
| *Crotaphytus bicinctores* | *cytb* | -0.857 | 0.177 | 0.120 | 1.000 |
| | *CARD4* | -1.087 | 0.143 | 0.120 | 0.400 |
| | *MC1R* | -0.460 | 0.365 | 0.103 | **0.000** |
| | *RAG1* | -1.399 | 0.074 | 0.094 | 0.094 |
| *Gambelia wislizenii* | *cytb* | -0.092 | 0.534 | 0.040 | 0.318 |
| | *CARD4* | -0.635 | 0.303 | 0.115 | 0.169 |
| | *MC1R* | -0.638 | 0.288 | 0.038 | 0.200 |
| | *RAG1* | -1.480 | 0.056 | 0.085 | **0.012** |
| *Dipsosaurus dorsalis* | *cytb* | 1.305 | 0.081 | 0.213 | 0.091 |
| | *CARD4* | 0.692 | 0.233 | 0.071 | 0.763 |
| | *MC1R* | -0.124 | 0.478 | 0.130 | 0.163 |
| | *RAG1* | 1.411 | 0.075 | 0.117 | **0.046** |
| *Sauromalus ater* | *cytb* | -0.134 | 0.515 | 0.526 | **0.005** |
| | *CARD4* | -0.551 | 0.340 | 0.121 | 0.999 |
| | *MC1R* | -0.976 | 0.186 | 0.170 | **0.002** |
| | *RAG1* | 0.077 | 0.415 | 0.085 | 0.412 |

Abbreviations: *cytb*, cytochrome b; CARD4, caspase recruitment domain gene 4; MC1R, melanocortin 1 receptor gene; RAG1, recombination activating gene 1; D, Tajima's D; R, Harpending's raggedness index. Significant p-values ($\alpha$ = 0.05) in bold.

Harpending's raggedness index results were inconsistent. All species examined had a significant R value at one or more loci ($\alpha$ = 0.05). This test assumes a null hypothesis of an expanding population, and a significant deviation from the null indicates a stable population size. Thus, as a test of population expansion, R is prone to type II errors, and the lack of consistency among the loci of each species may be the result of a lack of statistical power to detect a multipeaked distribution for some loci. Since at least one locus for each species had a significant R value, and no species had significant Tajima's D values, it seems reasonable to conclude that population sizes for these species have been essentially stable in recent time.

The ratio of synonymous to nonsynonymous base changes varied greatly among loci and species. For most species, the average across loci was consistently around 30% nonsynonymous (29.0% - 31.9%), but *D. dorsalis* had a much different ratio, with 61.1% of changes being nonsynonymous. When calculated by locus, it was found that the average proportion of nonsynonymous base changes across all species were 36.7% in CARD4, 23.3% in MC1R, and 58.0% in RAG1.

## Population subdivision

In order to test whether the high levels of genetic diversity seen in these species (particularly the crotaphytids) were simply the result of combining two genetically distinct subpopulations, we recalculated basic population genetic measurements for each putative subpopulation (defined by collection locality). Overall, haplotype diversities of the potential subpopulations were similar to those of the pooled samples for each species. Haplotype networks visually show the proportion of haplotypes which came from each collection locality (S1 Fig), and suggest very little population subdivision of haplotypes. *C. bicinctores* had the most differentiation, with significant differences in haplotype diversity between the two sampling locations at two loci (*cytb* and *CARD4*). Even at these two loci, however, one of the two subpopulations had a haplotype diversity value statistically indistinguishable from that of the total sample, showing that the high total diversity is not simply an additive effect. In addition, *C. bicinctores* had the

**Table 5. Measures of population subdivision.**

| Species | Gene | $F_{ST}$ | $\Phi_{ST}$ | Jost's D | $\Delta_{ST}$ |
|---------|------|----------|-------------|----------|---------------|
| *Crotaphytus bicinctores* | *cytb* | 0.249 | 0.177 | 1.000 | 2.000 |
| | *CARD4* | 0.061 | 0.107 | 0.136 | 1.082 |
| | *MC1R* | 0.037 | 0.172 | 0.692 | 1.580 |
| | *RAG1* | 0.011 | 0.083 | 0.044 | 1.039 |
| *Gambelia wislizenii* | *cytb* | 0.006 | 0.023 | 0.069 | 1.061 |
| | *CARD4* | 0.008 | 0.000 | 0.034 | 1.048 |
| | *MC1R* | 0.009 | 0.003 | 0.182 | 1.225 |
| | *RAG1* | 0.001 | 0.000 | 0.024 | 1.133 |
| *Dipsosaurus dorsalis* | *cytb* | 0.000 | 0.000 | 0.000 | 1.006 |
| | *CARD4* | 0.023 | 0.023 | 0.054 | 1.047 |
| | *MC1R* | 0.000 | 0.000 | 0.000 | 1.008 |
| | *RAG1* | 0.067 | 0.000 | 0.292 | 1.204 |

Abbreviations: *cytb* cytochrome b, *CARD4* caspase recruitment domain gene 4, *MC1R* melanocortin 1 receptor gene, *RAG1* recombination activating gene 1.

lowest diversity rank at *CARD4* despite the differences in haplotype diversity from the two sampling sites. $F_{ST}$ values ranged from 0.011 (*RAG1*) to 0.249 (*cytb*), and $\Phi_{ST}$ ranged from 0.083 (*RAG1*) to 0.177 (*cytb*) (Table 5). $\Phi_{ST}$ values were significantly different from zero at all loci in *C. bicinctores* ($\alpha$ = 0.05). Jost's D ranged from 0.044 (*RAG1*) to 1.000 (in *cytb*, where no haplotypes were shared between the two subpopulations). Effective number of subpopulations, $\Delta_{ST}$, ranged from 1.039 for *RAG1* to 2.000 for *cytb*.

In *D. dorsalis*, there were no differences among subpopulation and total haplotype diversity values at any locus. $F_{ST}$ values ranged from 0 (*cytb* and *MC1R*) to 0.067 (*RAG1*), and all $\Phi_{ST}$ values were statistically indistinguishable from zero. Jost's D ranged from 0 (*cytb* and *MC1R*) to 0.292 (*RAG1*). $\Delta_{ST}$ ranged from 1.006 for *cytb* to 1.204 for *RAG1*.

*G. wislizenii* samples also lacked any significant differences in haplotype diversity among subpopulations and the total. $F_{ST}$ values ranged from 0.001 (*RAG1*) to 0.009 (*MC1R*), and no $\Phi_{ST}$ values were significantly different from zero. Jost's D ranged from 0.024 (*RAG1*) to 0.182 (*MC1R*). $\Delta_{ST}$ ranged from 1.048 for *CARD4* to 1.225 for *MC1R*.

## Effect of trophic level/population size

Genetic diversity patterns between species differing in population size were nearly the opposite of what would be expected under neutral theory. Point estimates of the two small N species were ranked first and second most diverse for two of 4 genes (haplotype diversity) and for all 4 genes for $\Theta_S$. The average rank of the two small N species was greater than or equal to that of the large N species in all cases (Table 3). An alternative way of grouping the focal species is by habitat preference. Within a desert ecosystem, *D. dorsalis* and *G. wislizenii* are habitat generalists, and are often found in the open, while *C. bicinctores* and *S. ater* are saxicolous, found primarily among large rocks. Although rocky habitat is common at the study site, in general, it is a small fraction of the total habitat and we would expect habitat generalists to have larger overall populations than habitat specialists. The results do not show a consistent relationship between habitat specialization and genetic diversity (Table 3).

## Comparison with other species

We were able to compare haplotype diversity found in the present study with that found in other, more abundant lizard species from the same locality, sampled in previous studies (Fig 3;

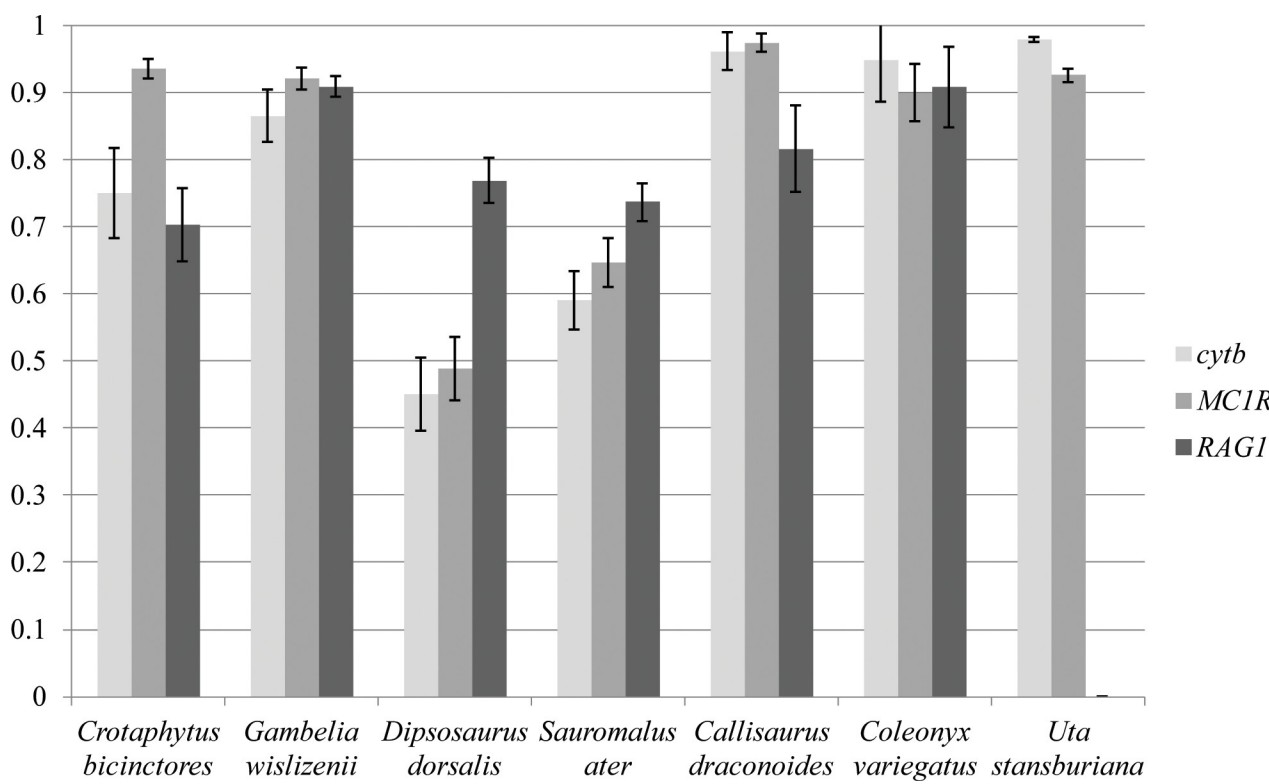

**Fig 3. Haplotype diversities of 4 focal species compared to more abundant lizard species from Mojave National Preserve.** Error bars represent ± 1 standard error. Abbreviations: cytb, cytochrome b; CARD4, caspase recruitment domain gene 4; MC1R, melanocortin 1 receptor gene; RAG1, recombination activating gene 1.

*CARD4* was not sequenced in previous studies [19,20]). To give an indication of the relative sizes of these species, we present the capture numbers from the same can trap study [35] cited above (*Callisaurus draconoides*—92, *Coleonyx variegatus* 108 and *Uta stansburiana*—1,529). The expectation was that the four larger bodied species studied here would have lower genetic diversity than all of the smaller, more common lizards studied. This was true for *cytb*, the only mitochondrial locus sequenced, where nearly all of three smaller species had significantly greater haplotype diversity than the four larger species examined here (the one exception was *C. variegatus*, with a haplotype diversity statistically indistinguishable from *G. wislizenii*).

This was not the case for the autosomal loci, however. The haplotype diversity of the two crotaphytids at *MC1R* was statistically indistinguishable from the values found in *U. stansburiana* and *C. variegatus*. At *RAG1*, levels of haplotype diversity in *C. bicinctores*, *D. dorsalis*, and *S. ater* are all indistinguishable from *C. draconoides* as well as one another (*RAG1* was not sequenced in *U. stansburiana*). Interestingly, haplotype diversity at this locus in *G. wislizenii* was significantly greater than that of *C. draconoides*, one of the most abundant lizards at this locality.

## Discussion

Our results do not conform to neutral expectations regarding the relationship between genetic diversity and population size. Among the four focal species examined for this paper, the species with the smallest population sizes (based on observation of current densities and the proxy criterion of trophic level) have diversity levels higher than those of species with larger

population sizes. The conclusion from comparing the 4 uncommon species surveyed for this study to the previously published results [19,20] is the same as the "matched comparison" within the 4 focal species of this paper. There is no clear relationship between the rank order of local population size. Indeed, these diversity levels are close to those of the lizard species with the very highest population sizes (Fig 3). A different population size proxy for the four study species would have been to categorize them as either habitat generalists (*Dipsosaurus dorsalis* and *Gambelia wislizenii*) or saxicolous (specialized for living around rocks; *Crotaphytus bicinctores* and *Sauromalus ater*). The generalist species would be predicted to have a larger population size, since more usable habitat is available to them, and therefore they should have higher genetic diversity under the neutral theory. However, the data show no evidence for this pattern either. The contrast in Table 3 between habitat generalists and specialists shows no consistent relationship across genes.

Although our results are more consistent with the predictions of selective models, there was no strong evidence for selection at the loci sequenced. This does not preclude the possibility that the diversity at these loci was shaped by selection at other, linked loci. The one exception was the gene *RAG1*, which may be under selection in these lizards. The highest proportion of nonsynonymous base changes was found at this locus, suggesting that there may be functional differences among the copies sequenced. Similar patterns were seen in *C. draconoides* and *C. variegatus*, where the percentage of nonsynonymous mutations ranged from 3.3–26.4% in *cytb* and *MC1R*, but were at 57.9–58.8% in *RAG1* [20].

We attempted to select markers that had as little linkage among loci as possible. Our autosomal loci BLAST to different chromosomes on the *Anolis carolinensis* genome (RAG1- Chromosome 1, CARD4—Chromosome 6, MC1R –unplaced genomic scaffold). If genetic draft were responsible for the patterns seen here, selective sweeps would likely have been pervasive, affecting much of the genome, rather than a few isolated events in the lineages studied. Interestingly, the variability at *cytb*, the mitochondrial locus, was not less than that of the autosomal loci. Unless there are great differences in mutation rate, it would be expected that genetic draft would have a greater effect on diversity of mtDNA because of the strong linkage disequilibrium among mitochondrial loci [7]. The mutation rate of mtDNA is expected to vary greatly among taxa [2] however, and we only sequenced one mitochondrial locus. Associative overdominance has been suggested as another mechanism by which genetic diversity in small populations may be greater than predicted by neutral theory [49].

Selection is not the only possible explanation for the lack of correlation between population size and genetic diversity in these lizard populations. Despite the fact that all the populations were sampled from the same small geographic area, it is possible that the different species experienced different phylogeographic or demographic events, such as vicariance or bottlenecks, or that they have very different mutation rates. However, Tajima's D showed no evidence of a bottleneck in any species. Pairwise mismatch analysis was unable to detect the multipeaked mismatch distribution characteristic of a stable population, but given our sample sizes this is likely to be lack of statistical power (Type II error) rather than positive evidence of a recent population expansion.

Our research was relatively limited in the number of species and loci surveyed, and the noise from coalescent variance(the among-locus variance in time to most recent common ancestor of a sample of DNA sequences from a locus) could mask the relationship between population size and genetic variation. An interesting next step for research with this study system would be to use next generation sequencing to sequence full genomes of all individuals. Full genome data would greatly increase the amount of data available to draw conclusions from, and provide several additional advantages. It would be possible to get a clearer view of how mitochondrial DNA (and other non-recombining segments like sex chromosomes) and

nuclear genomes differed in genetic diversity, and intraspecific variation in coding genes could be compared with that found in putative noncoding regions. It would also be possible to get good estimates of the effects of coalescent variance on the variation in genetic diversity among loci [50]. A recently published paper by Grundler *et al.* [51] took advantage of a long term census study of an Australian lizard community to examine the relationship between population size and genetic diversity using next generation sequencing data. They found a weak relationship between abundance and genetic diversity. They postulate that "nucleotide diversity is heavily influenced by factors other than census population size, or that ecological sampling in this community is unable to capture true population size." Interestingly, Grundler *et al.* found that genetic diversity was more strongly correlated with occupancy than with abundance. Assuming that occupancy is correlated with population connectivity, gene flow differences may be having an effect on genetic diversity by raising the effective population size for local population with higher gene flow rates. Our data are somewhat consistent with this idea, because *G. wislizenii* has the largest range of the four focal species and the lowest Fst values in our two location comparisons. As a habitat generalist, *G. wislizenii* may have a greater propensity for gene flow. Migration from outside populations may compensate for lower local populations sizes for some species. This may be why *G. wislizenii* has genetic diversities approaching those of the three species with much greater local population size (Fig 3), although gene flow is unlikely to explain the relatively high genetic diversity of *C. bicinctores*. As a rock specialist, we expect *C bicinctores* to have lower gene flow from neighboring populations because of intervening unsuitable habitat, and this is consistent with the higher Fst values for this species (Table 5).

Although our study does not by itself provide a definitive answer about the processes shaping genetic diversity, it is of interest because its results deviate from neutral expectations. It does not appear that population size has had a large effect on genetic diversity in this system, and that suggests that selection, demographic, and/or mutational differences may be playing an important role. If additional studies show a similar lack of relationship between population size and genetic diversity for matched sets of species, caution should be used when implementing algorithms to estimate population parameters like effective population size, time to most recent common ancestor, bottleneck effects, etc., as these methods assume relatively simple neutral models.

## Supporting information

**S1 Table. Specimen collection localities.** Collection locality for each lizard sampled, by species.
(DOCX)

**S2 Table. Haplotype randomization results.** Genetic diversity measures are shown for best supported pairs, along with the range of measures among simulated datasets.
(XLSX)

**S1 Fig. Haplotype networks.** Haplotype networks for all species, showing the haplotypes by collection locality. Within each network, circles are proportionate to number of copies of a haplotype (networks not on same scale). Hash marks between haplotypes represent the number of mutational steps.
(PDF)

**S1 File. Supplementary methods–Primers and PCR protocols.**
(DOCX)

**S2 File. R script for haplotype randomizations.**
(DOCX)

**S3 File. R script for calculation of $F_{ST}$ and Jost's D.**
(DOCX)

## Acknowledgments

Zarina Sheikh and other biology students at San Francisco State University participated in fieldwork and specimen collection. Previous graduate students in Eric Routman's lab, Mike Hague and Steven Micheletti, contributed to sample collection and produced some sequence data used in comparison with the present study.

## Author Contributions

**Conceptualization:** Erica M. Rutherford, Eric J. Routman.

**Data curation:** Erica M. Rutherford, Camille Kantor, Eric J. Routman.

**Formal analysis:** Erica M. Rutherford, Andrew Ontano, Camille Kantor, Eric J. Routman.

**Funding acquisition:** Erica M. Rutherford, Eric J. Routman.

**Investigation:** Erica M. Rutherford, Camille Kantor, Eric J. Routman.

**Methodology:** Erica M. Rutherford, Eric J. Routman.

**Project administration:** Erica M. Rutherford, Eric J. Routman.

**Resources:** Eric J. Routman.

**Software:** Erica M. Rutherford, Eric J. Routman.

**Supervision:** Eric J. Routman.

**Validation:** Erica M. Rutherford, Eric J. Routman.

**Visualization:** Erica M. Rutherford, Eric J. Routman.

**Writing – original draft:** Erica M. Rutherford, Eric J. Routman.

**Writing – review & editing:** Erica M. Rutherford, Andrew Ontano, Camille Kantor, Eric J. Routman.

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
