## [Decision Letter · Decision Letter 0]

27 Jun 2019

PONE-D-19-14658

Genetic variation across trophic levels: a test of the correlation between population size and genetic diversity in sympatric desert lizards

PLOS ONE

Dear Dr. Routman,

Thank you for submitting your manuscript to PLOS ONE. After careful consideration, we feel that it has merit but does not fully meet PLOS ONE’s publication criteria as it currently stands. Therefore, we invite you to submit a revised version of the manuscript that addresses the points raised during the review process.

I have now received reviews from two reviewers. The comments are very positive from the reviewers and I agree with them. However, there are several valid concerns raised by the reviewers, and I would suggest you to address them thoroughly before we can accept the manuscript. Therefore my decision is a major revision.

I agree with the reviewers that the manuscript needs a  (i) map with details of sampling locations and habitat types, (ii) explicitly stated hypothesis and predictions of this manuscript, (iii) changing some tables to figures for better presentation of the results, and (iv) better presentation of abundance and habitat specialization proxies. In addition, I would suggest to clearly and explicitly state the expectations as mentioned by one of the reviewers. For example, make a statement at the end of the introduction that “We tested neutral vs selection theories for relationship between genetic diversity and (a) body size, (b) population size, (c) trophic level, and (d) habitat specialization. Our expectation is that ….. under population size, genetic diversity would be …Carnivore specialist< carnivore generalist< herbivore specialist< herbivore generalist. “

I suggest to refrain from reference to ‘our lab’ and replace it with our previous research (citation) or previous research (citation). Also, please be consistent in how you use and present the species names: and *Genus species, or G species, or G.species, or Gs*: choose one and be consistent with it please.  

Congratulations on the positive reviews, I hope you will be to address the issues highlighted by the reviewers. Details on submitting the revisions are enclosed.

We would appreciate receiving your revised manuscript by Aug 11 2019 11:59PM. To enhance the reproducibility of your results, we recommend that if applicable you deposit your laboratory protocols in protocols.io, where a protocol can be assigned its own identifier (DOI) such that it can be cited independently in the future. For instructions see: http://journals.plos.org/plosone/s/submission-guidelines#loc-laboratory-protocols

We look forward to receiving your revised manuscript.

Kind regards,

Trishna Dutta

Academic Editor

PLOS ONE

Journal Requirements:

2. We noted in your submission details that a portion of your manuscript may have been presented or published elsewhere. Please clarify whether this publication was peer-reviewed and formally published. If this work was previously peer-reviewed and published, in the cover letter please provide the reason that this work does not constitute dual publication and should be included in the current manuscript.

Reviewers' comments:

Reviewer's Responses to Questions

**Comments to the Author**

1. Is the manuscript technically sound, and do the data support the conclusions?

Reviewer #1: Partly

Reviewer #2: Partly

2. Has the statistical analysis been performed appropriately and rigorously? 

Reviewer #1: Yes

Reviewer #2: Yes

3. Have the authors made all data underlying the findings in their manuscript fully available?

Reviewer #1: Yes

Reviewer #2: Yes

4. Is the manuscript presented in an intelligible fashion and written in standard English?

Reviewer #1: Yes

Reviewer #2: Yes

5. Review Comments to the Author

Reviewer #1: This is an interesting study that explores the relationship between population size and genetic diversity with four sympatric species- two herbivorous and two carnivorous lizards. I have a few broad and some specific comments.

Introduction

Generally, the introduction seems to be guided by the results that the authors got. If the motivation behind the study was to test whether neutral or selective forces shape genetic variation, this system and study design may not be most suited to address that.

In the introduction, the authors could focus more on the factors affecting genetic diversity in natural populations. It would be useful to have more information about both the intrinsic and extrinsic factors that affect genetic diversity. Authors mention the intrinsic factors of body size, mutation rates, generation times, etc. But extrinsic factors, such as extent and continuity of available habitat, do not find any mention.

66-67 Please elaborate on/ give examples of the confounding variables that you mention

Methods

91-92 It would be good to mention the body sizes here. People unfamiliar with the species and the region might find it useful.

Results

It would be nice to have a map showing study area with sample locations of the different species in different colours

It would be nice to compare generalist and specialist species within a trophic level instead of pooling them across the two levels as done in table 3.

A lot of information in the first paragraph under the ‘Mitochondrial DNA’ subheading can be moved to methods.

283-288 This information should go under the collection section in methods. Please also mention the distance between the different collection sites. Why was only one species sampled from Zzyzx? Other species do not occur in that area or were they not sampled for a reason?

Discussion

361-363 ‘The generalist species would be predicted to have a larger population size, since more usable habitat is available to them, and therefore they should have higher genetic diversity under the neutral theory. However, the data show no evidence for this pattern either.’ The authors cannot ignore the trophic level while making this statement. It is not okay to club the carnivore and herbivore generalists since population size is impacted by a combination of both- specialization and trophic level. Does the statement imply that the generalist herbivore in their study had a larger population size than the specialist carnivore? If yes, then its okay to club them in the same category and comment on expectations based on neutral theory. Whereas if they expect the population size in the following order: Carnivore specialist< carnivore generalist< herbivore specialist< herbivore generalist, the above statement cannot be made. The authors say nothing about the order in which they expect the population sizes to be for their study species. It should be explicitly mentioned in the introduction.

G. wislizenii has the lowest genetic differentiation and the highest genetic diversity based on multiple estimators. Does it also, among the study species, have a more continuous distribution and larger range?

Grundler et al (2019) does not find a reference in your paper. This recent study looks at the relationship between genomic diversity and abundance, occupancy and habitat specialization using genomic data from 30 species of lizards from arid Australia.

Reviewer #2: This study presents valuable data on genetic diversity, with an interesting ecological component that seeks to shed light on the still equivocal relationship between heterozygosity and population size. As such, it represents an important contribution to the field. However, there are several major points that I feel need reworking. My recommendation is publication after major revision, which should address the following issues:

1. The bulk of the data are presented in tables, which in my opinion lessens both the digestibility and impact of the results. The single figure that is presented could be improved with a key that indicates population size category of each of the species, such that readers can immediately pair heterozygosity with abundance/trophic category.

2. There are issues with the presentation of abundance data, which I feel impedes reproducibility of the study. The abundance data on which the authors base population size classification include a census study that I was unable to access; observational anecdotes (albeit over a long period of time); and coarse proxies. Proxies can be both necessary and valuable, but should be supported by more data or literature reference than found here. Similarly, observations of abundance should be supported with stronger justification as is done in Hague and Routman (2016) (frequently referenced in the current manuscript).

3. Because of the above two issues, I found it necessary to read Hague and Routman (2016) in conjunction with the manuscript in question. Much of the data and ecological context for the current study is presented in the 2016 manuscript. The two studies together present an interesting picture of the ambiguity of the relationship between genetic diversity and population size, and highlight the need for further study of ecological traits and genetic diversity. The current study seems to be a logical continuation of the 2016 project, and Figure 1 is a good example of the integration of data from each of these projects. However, the current manuscript could be improved by elaborating on the combined results in the discussion.

Specific comments in attached document.

6. PLOS authors have the option to publish the peer review history of their article (what does this mean?). If published, this will include your full peer review and any attached files.

Reviewer #1: No

Reviewer #2: No

---

## [Author Response · Author response to Decision Letter 0]

2 Sep 2019

28 August 2019

Dear Dr. Dutta:

 We would like to thank you and the reviewers for your careful review of our manuscript. We have revised the manuscript incorporating most of the comments from the reviews. This cover letter details the responses, organized by reviewer. The reviewers’ comments are in quotes and smaller font, followed by our response.

Reviewer 1

 “Generally, the introduction seems to be guided by the results that the authors got. If the motivation behind the study was to test whether neutral or selective forces shape genetic variation, this system and study design may not be most suited to address that.”

 This is the original motivation for this study, which is part of a larger study on multiple species in the same area. This study adds to work we have previously published (Micheletti et al. 2012, Hague and Routman 2016) that examines genetic diversity in sympatric populations of lizards. We are a Master’s only institution, and to accommodate the shorter time period that those students matriculate, individual projects need to be smaller than one would expect from a Ph.D. institution. We feel that the total system is actually quite appropriate for this question, but of course the comparison of genetic diversity to population size is only part of testing neutrality vs. selection, and we anticipate using next generation sequencing to test for the fingerprint of selection in future work. It should be noted that the main stimulus for theorists to develop selective theories for DNA sequence diversity is the supposed lack of correlation, predicted by neutral theory, between population size and genetic diversity when comparing taxa from very different parts of the tree of life. We are simply arguing that comparison of populations of closely related species from identical areas would be a better test of neutral theory. Such comparisons should be done by many different labs for taxonomically divergent groups before a definitive conclusion can be drawn about genetic diversity vs. population size.

“Authors mention the intrinsic factors of body size, mutation rates, generation times, etc. But extrinsic factors, such as extent and continuity of available habitat, do not find any mention.”

“66-67 Please elaborate on/ give examples of the confounding variables that you mention”

 The paragraph the reviewer is referring to is from Line 65-70. This is our attempt to justify 1) using taxonomically related species, and 2) using sympatric species, based on the criticisms from the previous paragraphs. We suggest that “focusing on a taxonomically related group” eliminates confounding variables that the reviewer correctly identifies as “intrinsic”. But we also suggest that studying species from a single area reduces some “extrinsic” forces like different geological histories or levels of gene flow (lines 69-71). We have altered the language to make this clearer. 

“91-92 It would be good to mention the body sizes here. People unfamiliar with the species and the region might find it useful. “

 We have added body sizes here.

 

 “It would be nice to have a map showing study area with sample locations of the different species in different colours.”

 We have added a map to show the desert region (satellite view) and the collection localities of each species. 

“It would be nice to compare generalist and specialist species within a trophic level instead of pooling them across the two levels as done in table 3.”

 This comparison can be made from Table 3, but we have made it more explicit by showing the results for each species/gene combination as a Figure. (We agree with all the reviewers that showing the results as only Table 3 is not the best way to convey the comparisons.). The pooled comparison is to show that even if we are wrong about the relative population sizes and both of the habitat specialists have lower population sizes, population size still does not explain the differences in genetic diversity. 

“A lot of information in the first paragraph under the ‘Mitochondrial DNA’ subheading can be moved to methods”

 We have moved some of this info to Methods, leaving only aspects which were done because of the specific results that were obtained.

“283-288 This information should go under the collection section in methods. Please also mention the distance between the different collection sites. Why was only one species sampled from Zzyzx? Other species do not occur in that area or were they not sampled for a reason?”

 We only collected Crotaphytus from Zzyzx because the species is uncommon enough that we had to collect from a different locality to get a sample size that approached that of the two herbivorous species. Gambelia was also less common than the herbivorous species, but is not common at Zzyzx. That is why we collected Gambelia at Kelso Dunes to help fill out this sample. When the study started, we were expecting that Gambelia and Crotaphytus would be less variable than the more common herbivorous species, even when combining samples from different areas, and therefore the combined samples would be a conservative test. This turned out not to be true, so we needed to do the analysis with and without combined samples, as described in the manuscript in the Results, subsection Population subdivision.

“361-363 ‘The generalist species would be predicted to have a larger population size, since more usable habitat is available to them, and therefore they should have higher genetic diversity under the neutral theory. However, the data show no evidence for this pattern either.’ The authors cannot ignore the trophic level while making this statement. It is not okay to club the carnivore and herbivore generalists since population size is impacted by a combination of both- specialization and trophic level. Does the statement imply that the generalist herbivore in their study had a larger population size than the specialist carnivore? If yes, then its okay to club them in the same category and comment on expectations based on neutral theory. Whereas if they expect the population size in the following order: Carnivore specialist< carnivore generalist< herbivore specialist< herbivore generalist, the above statement cannot be made. The authors say nothing about the order in which they expect the population sizes to be for their study species. It should be explicitly mentioned in the introduction.”

 We have rewritten this to be clearer that we were referring to a possible alternative contrast based on habitat specialization only, not the original hypothesis that carnivores are less common than herbivores and that within this contrast, habitat specialists are rarer than generalists. That is, we did not get the result that the observed-to-be-rarer species, the carnivorous lizards, were less diverse than the more common, herbivorous species. So one way to get our unexpected result would be if the habitat specialists as a group were actually less common than the habitat generalists. But that contrast did not match neutral theory explanations either. In fact, our genetic diversity results are not consistent with any contrast based on proxies for population size and neutral theory. 

“G. wislizenii has the lowest genetic differentiation and the highest genetic diversity based on multiple estimators. Does it also, among the study species, have a more continuous distribution and larger range?”

 We have included additional discussion of the possible role of gene flow in the Discussion. G. wislizenii does have a larger range than the three other focal species, although this may be an artifact of lower taxonomic/biogeographic attention that this species has received. For example, a mtDNA study (COIII) by Orange et al showed that the Mojave/Great Basin/Colorado plateau populations are separated from the Chihuahuan Desert populations by 6.1% sequence divergence. This suggests that gene flow from the Chihuahuan populations is unlikely to be influencing diversity in the more western populations. If the Chihuahuan Desert populations were designated as a separate species, there would not be as much difference in range sizes. The difficulty is that funding thorough multi-locus biogeographic studies is not a priority, so these studies may never be done.

 “Grundler et al (2019) does not find a reference in your paper. This recent study looks at the relationship between genomic diversity and abundance, occupancy and habitat specialization using genomic data from 30 species of lizards from arid Australia. “

 I only became aware of the Grundler et al. paper after we submitted the manuscript. We have incorporated some discussion of their paper. 

Reviewer 2

“The bulk of the data are presented in tables, which in my opinion lessens both the digestibility and impact of the results. The single figure that is presented could be improved with a key that indicates population size category of each of the species, such that readers can immediately pair heterozygosity with abundance/trophic category”

 We have added figures to correct this, as stated under “Reviewer 1”

“There are issues with the presentation of abundance data, which I feel impedes reproducibility of the study. The abundance data on which the authors base population size classification include a census study that I was unable to access; observational anecdotes (albeit over a long period of time); and coarse proxies. Proxies can be both necessary and valuable, but should be supported by more data or literature reference than found here. Similarly, observations of abundance should be supported with stronger justification as is done in Hague and Routman (2016) (frequently referenced in the current manuscript).”

 The abundance data are not great. The published survey (Persons and Nowak, 2007) is mostly consistent with my 20+ years of observation at this locality, but they did not show some very abundant taxa, such as Coleonyx variegatus, as being among the most common species, suggesting a collecting method bias or year to year variation in apparent abundance. We found a publication that includes the Wallace can trap data from the Zzyzx area, and we have added that to our revision. However, can traps are not effective at collecting Sauromalus, so the data include only 2 specimens of this species. We have added discussion of the fact that it took 4 years to collect the carnivore samples and much of that collecting was focused specifically on the carnivore species, whereas the herbivore sampling took less than two years and was done early in the study, when we were collecting all species of lizards and therefore not always in the best microhabitats for the two herbivore species that are the focus of this manuscript.

“Because of the above two issues, I found it necessary to read Hague and Routman (2016) in conjunction with the manuscript in question. Much of the data and ecological context for the current study is presented in the 2016 manuscript. The two studies together present an interesting picture of the ambiguity of the relationship between genetic diversity and population size, and highlight the need for further study of ecological traits and genetic diversity. The current study seems to be a logical continuation of the 2016 project, and Figure 1 is a good example of the integration of data from each of these projects. However, the current manuscript could be improved by elaborating on the combined results in the discussion.” 

 We have elaborated on the combined results in the discussion.

“L41-43 – what are the hypotheses for deviations from expectation in these studies? Good to have context here about why we might not expect things to correlate”

 We added a statement that the authors credit natural selection for the deviations from neutral expectations. However, the possibility of coalescent variation (variation among loci in time to most recent common ancestor, which can vary enormously even for different loci sampled from exactly the same population) and variation in mutation rates is also present.

L59 – not sure about the use of “haphazard” here; larger scale analyses are important for other reasons, especially considering impacts of population connectivity on genetic diversity

 We eliminated the term haphazard.

L117-118 – references for this?

 We have added two references for this statement.

L118-120 – References for census studies are inaccessible. I feel more detail is needed here to justify the validity of qualitative data – observations on abundance across seasons? Over periods of how long? Etc. Even a figure plotting the cited census data with genetic diversity would be valuable.

 We have added more information about our justification for relative population sizes, including another reference to Wallace’s can trap study. [Cummings KL, Puffer SR, Holmen JB, Wallace JK, Lovich JE, Meyer-Wilkins K. Biodiversity of amphibians and reptiles at the Camp Cady Wildlife Area, Mojave Desert, California and comparisons with other desert locations. California Fish and Game 2018 Summer 104(3): 129-147].

L124 – how large was the main collection area, and what are overall geographic ranges like for these species? More detail on connectivity of these species with surrounding populations would be useful when considering genetic diversity and population size (especially when there are deviations from expectation as we have here) 

 We have no data on the connectivity among surrounding populations (although we do Fst analysis of the two species with samples from multiple locations. As noted above, we have added a discussion of the possible role of gene flow to the Discussion.

L147-174 – good use of alternate tests and simulations to validate analyses

 Thanks.

L160-162 – there are standard statistical tests to quantify significant difference in pairwise measures that are sounder than a decision made by visual inspection. The citation for this method comes from a psychology article that I am unable to access, and seems, based on the abstract, more relevant for gaining inference from figures and improving research communication in a field that operates under different methods standards. 

 The standard error approach is not simply visual, and involves a calculation of the overlap. However, it does make assumptions about the shape of the credible interval, especially that the variation is normally distributed. We have calculated the t test suggested by Masatoshi Nei for the haplotype diversity contrasts, and found some additional significant differences. However, because the direction of the difference in diversity is counter to the predictions of neutral theory, our original method is conservative. That is, discovering additional statistical significance does not change the conclusion that the diversities do not match those predicted by neutral theory. 

L187:

 Table 1 - Dn/ds statistic could be presented as a single column rather than or in addition to #synonymous and #nonsynonymous mutations in separate columns (which makes interpretation of this metric a two-step rather than one-step process)

 Done

Tables summarized into figures would be helpful, especially Table 3. Much of this appears in fig 1, but the summary of information could be improved; cannot immediately tell which are small pops, which are large pops, which trophic level the species represent, etc. It’s not a lot to keep track of or to find in the text, but figures should be interpretable by themselves.

 Done (see response to Reviewer 1 for the same criticism).

“L236 - Tables 4 and 5 are identical”

 Oops! The correct Table 5 has been added.

“L334 – references for eco data? Found in open habitat doesn't necessarily = generalist…if rocky habitat is common then rock specialists could also be common…”small fraction of total habitat” – any kind of quantification available? What is the rest of the habitat characterized by? Ecological specialization is a very interesting and important consideration, and a valuable addition to the study, but it does feel a little too coarse as it stands. It would be helpful to at least include some references to strengthen inferences made here. “

 We have added language to our discussion of the collection maps to discuss the fact that the area of suitable habitat for the rock dwelling species is less than that of the species that prefer the open areas. In addition, the habitat generalists also make use of the rock areas in addition to the open areas. Habitat specialization is simply more justification for our ranking of relative population sizes. (I am aware of no population genetics theory that relates the degree of habitat specialization to genetic diversity, other than through its effect on population size).

L386-387 - Reasoning behind “likely lack of statistical power”? Authors have a good number of individuals - do they mean due to number of species? Could use a little clarification

 This is not a “number of species” problem. We discuss this a bit in the results section (Lines 270-276 of the original manuscript). This test relies on the ability to differentiate a multi-modal distribution (of pairwise nucleotide mismatches among sampled copies of a locus) from a unimodal distribution. Differentiating distributions require a large number of sampled copies to have strong statistical power. In this case we are trying to infer a population expansion by detecting a unimodal distribution. In the statistical test, the unimodal distribution is the null hypothesis, so that failure to reject the null could be due to population expansion or lack of power.

L395 – need some background here to contextualize coalescent variance 

 Coalescent variance is the term to describe the variation in TMRCA (Time to Most Recent Common Ancestor) among loci sampled from a common ancestor. As a result, it follows that there can be differences in genetic diversity of a given locus sampled from different populations/species just because the locus has a recent TMRCA in one population and a more ancient TMRCA in another population, simply by chance. I believe that the term “coalescent variance” is in common use in population genetics, but we can include a paragraph like the one in this response if the editor feels that it would be useful. For this revision we have defined it in a few words in the text on its first mention (which is L389 in the original manuscript) as “coalescence variation”, which I have changed to “coalescent variance” for consistency.

L407 – It would be valuable to situate the results of this study better in the context of previous results – Hague and Routman (2016) showed clear relationship between diversity and pop size but now we have conflicting relationship when we throw smaller pops into the mix – what could be going on here? This is especially interesting and merits additional study with all species to look for ecological correlations – and so should be mentioned with more detail in the discussion

 We have elaborated on the discussion to discuss the other species for which we have data and the paper mentioned by reviewer 1.

Associate Editor Dutta

 “I agree with the reviewers that the manuscript needs a (i) map with details of sampling locations and habitat types, (ii) explicitly stated hypothesis and predictions of this manuscript, (iii) changing some tables to figures for better presentation of the results, and (iv) better presentation of abundance and habitat specialization proxies. In addition, I would suggest to clearly and explicitly state the expectations as mentioned by one of the reviewers. For example, make a statement at the end of the introduction that “We tested neutral vs selection theories for relationship between genetic diversity and (a) body size, (b) population size, (c) trophic level, and (d) habitat specialization. Our expectation is that ….. under population size, genetic diversity would be …Carnivore specialist< carnivore generalist< herbivore specialist< herbivore generalist”

 Items i, iii, and iv are address in responses to Reviewers 1 and 2. We have added language to address explicit hypotheses and expectations.

“I suggest to refrain from reference to ‘our lab’ and replace it with our previous research (citation) or previous research (citation).”

 Done.

“Also, please be consistent in how you use and present the species names: and Genus species, or G species, or G.species, or Gs: choose one and be consistent with it please.”

 We have corrected any deviations from the traditional usage of scientific names: the first use of the name is written out fully and thereafter the genus name is abbreviated as a single letter and only the species name is written out fully. The exception is for readability of the new figures: we adopted the use of a 4 letter code for the species that consists of the first two letters of the genus and species names, e.g. CRBI for Crotaphytus bicinctores. This is defined in the figure legends. If the editor would like us to be consistent across text and figures, we could easily adopt the 4 letter codes in the text as well.

Sincerely,

Eric Routman 

References:

Hague MTJ, Routman EJ. Does population size affect genetic diversity? A test with sympatric lizard species. Heredity. 2016 Aug 26;116:92-98.

Micheletti S, Parra E, Routman EJ. Adaptive color polymorphism and unusually high local genetic diversity in the Side-Blotched Lizard, Uta stansburiana. PLoS One. 2012 Oct 25;7(10):e47694.

---

## [Editor Report · Decision Letter 1]

4 Oct 2019

Genetic variation across trophic levels: a test of the correlation between population size and genetic diversity in sympatric desert lizards

PONE-D-19-14658R1

Dear Dr. Routman,

We are pleased to inform you that your manuscript has been judged scientifically suitable for publication and will be formally accepted for publication once it complies with all outstanding technical requirements.

With kind regards,

Trishna Dutta

Academic Editor

PLOS ONE

Additional Editor Comments (optional):

Dear Dr. Routman,

Congratulations on submitting a well-addressed reply answering the issues raised by the two reviewers and myself. I find that you have addressed a majority of the issues satisfactorily, and I am happy to accept this manuscript for publication in PLOSONE.

My only minor comments are: 

(1) I encourage you to insert an inset in Fig1 (map of the study area), and adjust marker transparency so that the underlying collection locations can be visualized.

(2) Page 19, L 368: First word should be Indeed, but is ndeed. Please correct this.

Thank you and Congratulations

Regards,

Trishna Dutta
---

## [Editor Report · Acceptance letter]

22 Nov 2019

PONE-D-19-14658R1 

Genetic variation across trophic levels: a test of the correlation between population size and genetic diversity in sympatric desert lizards 

Dear Dr. Routman:

I am pleased to inform you that your manuscript has been deemed suitable for publication in PLOS ONE. Congratulations! Your manuscript is now with our production department. 

With kind regards,

on behalf of

Dr. Trishna Dutta 

Academic Editor

PLOS ONE